# Regulation of Heat Stress in *Physcomitrium* (*Physcomitrella*) *patens* Provides Novel Insight into the Functions of Plant RNase H1s

**DOI:** 10.3390/ijms23169270

**Published:** 2022-08-17

**Authors:** Zhuo Yang, Liu Duan, Hongyu Li, Ting Tang, Liuzhu Chen, Keming Hu, Hong Yang, Li Liu

**Affiliations:** 1State Key Laboratory of Biocatalysis and Enzyme Engineering, Hubei Collaborative Innovation Center for Green Transformation of Bio-Resources, Hubei Key Laboratory of Industrial Biotechnology, School of Life Sciences, Hubei University, Wuhan 430062, China; 2School of Life and Health Sciences, Hunan University of Science and Technology, Xiangtan 411201, China; 3Jiangsu Key Laboratory of Crop Genomics and Molecular Breeding/Key Laboratory of Plant Functional Genomics of the Ministry of Education, College of Agriculture, Yangzhou University, Yangzhou 225009, China; 4Co-Innovation Center for Modern Production Technology of Grain Crops of Jiangsu Province/Key Laboratory of Crop Genetics and Physiology of Jiangsu Province, Yangzhou University, Yangzhou 225009, China

**Keywords:** PpRNH1A, heat stress (HS), lipid droplets, *Physcomitrium* (*Physcomitrella*) *patens*

## Abstract

RNase H1s are associated with growth and development in both plants and animals, while the roles of RNase H1s in bryophytes have been rarely reported. Our previous data found that *PpRNH1A,* a member of the *RNase H1* family, could regulate the development of *Physcomitrium (Physcomitrella) patens* by regulating the auxin. In this study, we further investigated the biological functions of *PpRNH1A* and found *PpRNH1A* may participate in response to heat stress by affecting the numbers and the mobilization of lipid droplets and regulating the expression of heat-related genes. The expression level of *PpRNH1A* was induced by heat stress (HS), and we found that the *PpRNH1A* overexpression plants (*A*-OE) were more sensitive to HS. At the same time, *A*-OE plants have a higher number of lipid droplets but with less mobility in cells. Consistent with the HS sensitivity phenotype in *A*-OE plants, transcriptomic analysis results indicated that *PpRNH1A* is involved in the regulation of expression of heat-related genes such as *DNAJ* and *DNAJC*. Taken together, these results provide novel insight into the functions of RNase H1s.

## 1. Introduction

With global warming, heat stress (HS) has gradually become a major limiting factor affecting plant growth, plant geographic distribution, crop yield, and quality [1,2,3]. Research analysis showed that the negative impact of high temperature on crop yield is becoming more severe [4]. Plants do not move and have evolved a series of self-protective mechanisms against external stresses to adapt to the environment [5]. Heat shock proteins (HSPs) and molecular chaperones are commonly found to be involved in plants’ responses to HS [6,7]. DnaJ (HSP40) proteins are important molecular chaperones involved in signal transduction, cellular proteostasis, and tolerance to various stresses in plants [8,9]. The expression levels of these genes are up-regulated to improve heat tolerance in plants [4,6]. For example, *GmDNJ1* (a major *HSP40*) is highly induced at high temperature, and *Gmdnj1*-knockout mutants have more severe browning and lower chlorophyll content, and higher reactive oxygen species (ROS) content under HS, which suggest that *GmDNJ1* plays an important role in response to heat stress in soybean [10].

Cytosolic lipid droplets play important roles in plant growth and responses to stress. Lipid droplets (or so-called lipid bodies in some cases) are subcellular structures that are capable of storing neutral lipids and various hydrophobic compounds in eukaryotes, which are present in almost all green plant lineages and is widely explored in various kind of tissues such as seeds, pollen, and leaves in different plant species [11,12,13,14]. Feeney et al. found that a large ectopic accumulation of lipid droplets in vegetative organs produced abnormal embryogenic structures in plants [15]. Triacylglycerol (TAG) is one of the main components of lipid droplets. Abnormally accumulating TAG in plant leaves usually affects plant growth and causes cell death [15], which was also observed when subjected to various stresses [16]. Abiotic stresses such as heat stress, high light, drought, and cold induce lipid droplets accumulation in plants [17], including not only land plants but also chlorophyte algae such as *Haematococcus pluvialis*, *Chlamydomonas reinhardtii*, and cyanobacterium *Synechocystis* sp. PCC6803, among others [18,19,20]. Furthermore, the accumulation and mobilization of lipid droplets were reported to be positively related to plant stress resistance [17,21,22]. In the salt-sensitive variety, salt stress resulted in lipid droplets accumulation and higher lipid droplets retention, whereas the tolerant variety exhibited faster lipid droplets mobilization [22]. In *P. patens*, lipid droplets were found to be present in dehydrated spores and photosynthetic gametophytes [14]. Previous studies have shown that numerous lipid droplets could be observed after dehydration, which indicates that drought stress also induces the accumulation of lipid droplets in *P. patens* [23]. Lipidome analysis revealed that lipid metabolism plays an important role in *P. patens* to cope with the terrestrial environment stresses, although the composition of monogalactosyldiacylglycerol is different between *P. patens* and vascular plants [24].

RNase H is considered as a class of sequence-nonspecific ribonucleases and was first isolated and identified from calf mammary glands [25]. RNase Hs were found widespread in archaebacteria, prokaryotes, and eukaryotes [26] and are classified into three types (RNase H1, H2, and H3) [27]. Among them, RNase H3 only exists in certain archaebacteria [28,29], while RNase H1 and RNase H2, are present in plants and animals [30,31]. Although the structure of RNase H1s are evolutionarily highly conserved [32], RNase H1s were found to play important roles in diverse biological processes such as replication process, RNA processing process, and development of mitochondrial DNA [33,34]. In addition, they play crucial roles in the maintenance of genomic stability, the repair of DNA damage, and affect the development of organisms as well [35,36,37]. Three AtRNH1s were found in *Arabidopsis thaliana*, which are localized in the nucleus (AtRNH1A), mitochondria (AtRNH1B), and chloroplasts (AtRNH1C), respectively [38]. The diversification results of the subcellular localization of RNAH1s in *Arabidopsis* facilitate us to hypothesize that RNH1s may have multiple functions in different cellular compartments. In addition, AtRNH1B and AtRNH1C are required for viability in *Arabidopsis*, while AtRNH1A is not [39]. Moreover, studies showed that loss-of-function mutation of AtRNH1C exhibits a distinct growth phenotype of dwarfism and leaf chlorosis [39]. Gene functional mechanism studies showed that AtRNH1C and AtGyrases interact to reduce the damage of DNA [38], and AtRNH1C promotes the repair of DNA damage in chloroplasts mainly by its synergistic action with ssDNA-binding proteins (WHY1/3) and recombinases (RecA1) as well [40]. Interestingly, deletion of AtRNH1B allows AtRNH1C, which is located in chloroplasts, to enter the mitochondria, ensuring the integrity of the mitochondrial genome [41].

Despite functional research of RNH1s in *Arabidopsis* in recent years, it remains unclear how RNH1s function in response to abiotic stresses and how RNH1s evolved within land plant lineage. Mosses, which belong to the early land plants, hold an important position in higher plant evolution. *Physcomitrium patens* (*P. patens*), has been a model plant to study evolutionary developmental and stress tolerance/adaption questions as a non-seed plant [42,43,44]. Therefore, it is of some far-reaching significance to study the biological functions of the *RNase H1* family in *P. patens.* In our previous research, we found that there are two family members of *RNase H1s* in *P. patens* [45]. PpRNH1A affects shoot growth and branch formation of *P. patens* by controlling the formation of the R-loop and regulating the transcription level of auxin-related genes in the mutant *pprna1a* [45]. However, different from the subcellular location results of AtRNH1A, PpRNH1A was found localized in both nucleus and cytosol, which suggested a species-specific role of RNH1A in *P. patens*. In this research, we further explore the function of PpRNH1A and found that overexpression of *PpRNH1A* (*A*-OE) plants were more sensitive to heat stress by regulation of heat-related gene expressions, such as *DNAJ* and *DNAJC*. Chloroplast development but not pigment synthesis was affected by overexpression of *PpRNH1A.* Ultrastructural and surface structural observations show growth defects in *A*-OE plants, and abnormal accumulation and mobilization of lipid droplets were found in the cytosol, which is associated with the sensitivity of *A*-OE plants to heat stress.

## 2. Results

### 2.1. Phylogenetic Analysis of RNases H1s

In our previous study, we identified two family members of *RNase H1s* in *P. patent*, named PpRNH1A (Pp3c4_14290) and PpRNH1B (Pp3c2_25170), which contain the RNase H catalytic domain [45]. To investigate the evolutionary relationship of RNase H1s in plants, we used the full-length protein sequence of PpRNH1A from the Phytozome database to search for the RNase H1s homologs in representative species such as algae, mosses, and higher terrestrial plant groups. Public database Phytozome, Ensembl plants, NCBI, and the *klebsormidium* genome project were used for BLAST analysis. The phylogenetic tree (Figure 1) showed that three main branches were identified according to their full-length protein sequences, and the development process of RNase H1s was consistent with the evolutionary history of organisms. PpRNH1A and PpRNH1B were grouped together and showed a close evolutionary relationship to *Chlamydomonas reinhardii* and *Klebsormidium nitens*. One to three family members of RNase H1s were identified in different plant species, and some of the proteins in the same species were assigned to different sub-branches, which suggested that they may have differentiated functions.

### 2.2. PpRNH1A Is Involved in Growth Development and Stress Tolerance

To explore the features of the *PpRNH1A* (Pp3c4_14290), we first investigated the expression pattern of the *PpRNH1A* in wild type throughout the growth cycle of *P. patens*. Expression data were downloaded from the public database *Physcomitrella* eFP Browser (BAR, http://bar.utoronto.ca/, data shown in Figure 2 were accessed on 19 February 2022). The higher expression levels of *PpRNH1A* were found in spores, rhizoids, archegonia, and at the sporophyte S1 stage, with relatively lower expression levels in chloronema, gametophores, and at the S2, S3, and M stages of sporophyte development (Figure 2A,B). This result suggests that PpRNH1A might be involved in *P. patens* growth and development.

We next examined the expression levels of *PpRNH1A* under abiotic stresses. The wild-type gametophores of *P. patens* were exposed to heat stress (HS) at 40 °C for 18 h and then recovered for 4 days, and samples at three timepoints (control, heated, and recovered) were taken to determine the relative expression level of the *PpRNH1A*. Our results showed that HS significantly induced the expression of the *PpRNH1A* (3.91 times induction compared to the control) with a continuous induction (6.33 times induction compared to the control) during the recovery period (Figure 2C). Furthermore, *PpRNH1A* expression levels under dehydration stress treatment were extracted from our previous dehydrated transcriptome data [46]. The result showed that *PpRNH1A* was slightly induced (1.47 times induction compared to the control) by dehydration stress treatment and then returned to a normal level after recovery (Figure 2D). Our results showed that *PpRNH1A* participates in both development and stress tolerance in *P. patens*.

### 2.3. Creation of Stable Transgenic P. patens Line Overexpressing PpRNH1A

To detect the functions of *PpRNH1A*, we obtained the CDS of the *PpRNH1A* gene by PCR amplification. The CDS fragment without the stop codon was integrated into the overexpression vector (Figure 3A) under the promoter of *PpEF1a* followed by a PEG-mediated *P. patens* protoplast transformation. Positive *PpRNH1A*-overexpression plants were selected using the hygromycin resistant marker and confirmed at both DNA and RNA levels (Figure 3B,C). Seven positive transformants at the DNA level were obtained. RNA was taken from each line and followed by quantitative real-time polymerase chain reaction (qRT-PCR) to confirm the expression levels of *PpRNH1A*-overexpressing lines (*A*-OE). One *A*-OE with a three times overexpression level compared to wild-type plants was shown in Figure 3C. The pigment contents, including Chlorophyll A (Chl a), Chl b, total Chls, and carotenoids in WT and *PpRNH1A*-overexpression plants were measured under normal growth conditions, and the result showed that no significant changes were found between WT and *A*-OE in pigment contents (Figure 3D).

### 2.4. Overexpression of PpRNH1A Affects Lipid Droplets Metabolism and Mobilization

The intracellular ultrastructure of *A*-OE plants was observed by transmission electron microscopy (TEM). We found that the cell walls of the overexpression plants were much thinner than the WT (Appendix A). Interesting, there was no clear laminal structure of the chloroplast stroma observed in *A*-OE plants (Appendix A), although the level of pigments in *A*-OE seems not to have been affected. Furthermore, in *A*-OE plants, more plastoglobuli were observed in chloroplasts (Appendix A). The surface of the plant was then observed by scanning electron microscopy (SEM). The results showed that there were protrusions like glandular hairs on the surface of the *A*-OE plants (Appendix A). These results lead us to hypothesize that the lipid metabolite was affected in *A*-OE plants, since the plant cuticle is a lipid membrane covering plant surfaces, and the plastoglobuli is a kind of lipid droplets within the chloroplasts.

To confirm whether the overexpression plants could affect the accumulation of cytosolic lipid droplets, the dye Nile red was used to stain the *A*-OE plants and wild-type plants to observe the cytosolic lipid droplets under the confocal microscope. As shown in Figure 4, the yellow fluorescence represented the accumulation of cytosolic lipid droplets by Nile red. Chloroplast autofluorescence was marked with red fluorescence in both wild-type and *A*-OE plants (Figure 4). The results showed that the lipid droplets were hardly seen in the gametophore of wild-type plants under normal growth conditions, however, in contrast, abundant of cytosolic lipid droplets around chloroplasts were observed in *A*-OE plants (Figure 4). In addition, a proportion of the lipid droplets observed in the overexpression plants had irregular morphology (Figure 4). Although additional experimental evidence is needed, these results revealed that overexpression of *PpRNH1A* may be associated with the metabolism and mobilization of lipid droplets.

### 2.5. PpRNH1A-Overexpression Line Is More Sensitive to Heat Stress

To detect the function of the protein deduced by *PpRNH1A*, plants were exposed to heat stress. Thirty days old gametophores of the wild-type plants and *A*-OE plants were treated with 40 ℃ for 18 h. Followed by a recovery at normal temperature (25 °C) for 5 days. Results showed that *A*-OE plants were more sensitive to heat stress and could not recover from the HS (Figure 5A). Photosynthetic parameters, including the maximal efficiency of PSII photochemistry (Fv/Fm), non-photochemical energy dissipation (NPQ), and electron transport rate (ETR) were measured at the timepoints of before heat stress (the control), after heat stress, and recovered for 4 days. Similar Fv/Fm, NPQ, and ETR levels were observed between *A*-OE plants and WT before HS. Heat stress severely affects the function of the photosystem in both *A*-OE plants and WT, whereas the Fv/Fm and NPQ levels in *A*-OE plants were significantly lower than in WT plants, both under heat stress and after recovery (Figure 5B–D). These results revealed that the overexpression of *PpRNH1A* resulted in decreasing tolerance to heat stress in *P. patens* plants.

### 2.6. PpRNH1A Regulates the Expression of Heat-Responsive Genes including DNAJ and DNAJC

Considering that overexpression plants are more sensitive to heat stress, we then tested if *PpRNH1A* regulates the heat-related genes. Transcriptomic profiles of WT plants before HS (control) and 18 h after HS were compared to identify the heat-responsive genes. In total, 10,538 DEGs (Fold change > 1.5 or 0 < Fold change < 0.67, *p*-value < 0.05) were found responding to heat stress in wild-type plants (collection: WT-vs-WT_H). To identify the genes regulated by *PpRNH1A*, 3768 DEGs were identified from the comparison between WT and *A*-OE plants under normal growth conditions (collection: WT-vs-*A*-OE). The overlap of these two collections of DEGs showed that 2292 heat-related genes were regulated by *PpRNH1A* (Figure 5E). Representative heat-related genes [47] such as *Pp3c27_7640* (molecular chaperone *DNAJ*, *HSP40*) and *Pp3c21_19830* (DNAJ homolog subfamily C member *DNAJC*) were selected for further analysis. Heat stress significantly induced the expression level of *DNAJ* and *DNAJC* in WT plants. However, in *A*-OE plants, *DNAJC* was not induced by heat (Figure 5F). The fold change of *DNAJ* induced by HS in *A*-OE plants was much less than that in the WT plants. These results suggest that *PpRNH1A* regulates the expression of heat-responsive genes, including *DNAJ* and *DNAJC* (Figure 5F).

## 3. Discussion

RNase H1s are ribonucleases widely present in organisms [26], and most studies on RNase H1s are related to growth and development. In *Arabidopsis*, AtRNH1B and AtRNH1C play important roles in maintaining the normal development of embryos, and AtRNH1C deletion mutants showed severe growth-defective phenotypes [39]. Our recent progress on *P. patens* revealed that *PpRNH1A* regulates the transcription of auxin-related genes by controlling the formation of R-loops, further regulating the development [45]. In this study, we found novel biological functions of PpRNH1A, which may also participate in heat stress responses, possibly by affecting the numbers and the mobilization of lipid droplets and regulating the expression of heat-related genes.

Phylogeny analysis results in this study suggest that there is functional differentiation of RNase H1s within and among plant species (Figure 1), which confirmed our previous finding that there may be functional differentiation between AtRNH1A and PpRNH1A [45]. Previously, we found that the development of gametophores of the *pprnh1a* mutant was affected through the modulation of R-loop formation on auxin-related genes. Overexpression plants of *PpRNH1A* were obtained (Figure 3A) in this study, and we found that the chloroplasts development was affected and no clear lamellar structure was observed (Appendix A). Interestingly, although the chloroplast development was impaired, the pigments did not show any difference in *A*-OE plants compared to WT plants (Figure 3D). Abnormal development of lamellar structure and more plastoglobuli were found in the chloroplasts of the *A*-OE plants (Appendix A). Furthermore, protrusions like glandular hairs were found on the surface of the *A*-OE plants as well. In addition, abundant lipid droplets with irregular morphology were accumulated in the cytosol of *A*-OE plants (Figure 4). These results suggest that overexpression of *PpRNH1A* resulted in disordered lipid metabolism and mobilization.

TAGs are the primary constituents of lipid droplets, which do not accumulate in vegetative tissues of plants under normal growth conditions but accumulate significantly under stress conditions, such as drought, high temperature, low temperature, and nutrient starvation, especially in leaves [48,49]. Correspondingly, lipid droplets were reported to be closely associated with plant biotic and abiotic stress as well. *Pseudomonas*-infected leaves caused an accumulation of lipid droplets and induced hypersensitivity reactions in *Arabidopsis* [14]. Seeds with irregular lipid droplets morphology were found susceptible to chilling injury during germination [50,51]. Consistent with the above, we found that *PpRNH1A* is involved in plant responses to heat stress, which provided a novel function of RNase H1s in plants. In our study, the expression levels of *PpRNH1A* were found to be significantly induced by abiotic stresses such as heat (Figure 2C) and drought treatment (Figure 2D). Moreover, *A*-OE plants were sensitive to heat stress (Figure 5A), further confirmed by the levels of photosynthetic parameters (Figure 5B–D). This is coincident with the findings in sunflower (*Helianthus annuus* L.), where stress-sensitive lines showed longer retention of the lipid droplets membrane under salt stress, thus exhibiting higher lipid accumulation and faster mobilization than that of stress-tolerant lines [22]. As a note, the lipid droplets mentioned here in this study are not what had been described as “oil body” in liverworts. Lipid droplets are present in almost all green plants accumulating compounds that are not soluble in the aqueous phase, and oil bodies are present only in liverwort, storing large quantities of toxic sesquiterpenoids [17,52,53,54,55]. Lipid droplets are subcellular organelles of monolayer membranes with a diameter of about 0.5–2 µm, while oil bodies are an endocrine structure surrounded by a lipid bilayer membrane [13,55]. The size of small oil bodies is 2–5 × 3–9 µm on average, and the diameter of large oil bodies can reach 70 µm [56,57]. Although both mosses and liverworts are bryophytes, lipid droplets were found in our experiment.

HS affects all aspects of plant growth and elicits responses in a range of genes, including the accumulation of heat shock proteins (HSPs). We found that 60.8% of the genes regulated by overexpression of *PpRNH1A* are responsive to heat stress (Figure 5E), which is one of the explanations for why *A*-OE plants were more sensitive to HS than wild-type plants (Figure 5A–D). Heat responsive genes play critical roles in regulating the resistance/tolerance of plants to heat stress. *GmDNJ1* (a major *HSP40*) was induced by heat stress and was responsible for enhanced heat tolerance in soybean [10]. In tomatoes, the expression levels of *LeCDJ1* of *Lycopersicon esculentum* and *SlDnaJ20* of *Solanum lycopersicum* were induced by heat stress, and overexpression of *LeCDJ1* or *SlDnaJ20* could improve the heat tolerance of plants [9,58]. In alfalfa (*Medicago sativa*), the DnaJ-like protein (*MsDJLP*) gene was induced by heat stress, and ectopic expression of *MsDJLP* in tobacco enhances the heat tolerance of tobacco [59]. All of these indicate that DNAJ could be used as a representative heat responsive marker and its expression level is closely related to heat stress tolerant ability. Our results showed that both *DNAJ* and *DNAJC* were induced by heat stress in WT plants, however, on the contrary, the induction folds in the *A*-OE plants were not as strong as that in the wild type, especially *DNAJC,* whose induction was totally hampered under heat stress. This is corroborated by the result that our *A*-OE plants is more susceptible to heat stress.

## 4. Materials and Methods

### 4.1. Phylogenetic Analysis

Amino acid sequences used were obtained from Phytozome (https://phytozome.jgi.doe.gov/pz/portal.html, accessed on 19 February 2022), Ensembl plants (http://plants.ensembl.org/Cyanidioschyzon_merolae/Tools/Blast, accessed on 19 February 2022) and NCBI (https://www.ncbi.nlm.nih.gov, accessed on 19 February 2022) database. And the RNase H1 of the *Klebsormidium nitens* was obtained from the *klebsormidium* genome project website (http://www.plantmorphogenesis.bio.titech.ac.jp/~algae_genome_project/klebsormidium/index.html, accessed on 19 February 2022) [60]. Multiple sequence alignments of these amino acid sequences were conducted with the ClustalW of MEGA-X, and the phylogenetic tree construction was performed with MEGA-X using the maximum likelihood (ML) method and 1000 bootstrap [61]. The best-fit model we selected was JTT + G + I [62].

### 4.2. Plant Materials and Growth Conditions

Gransden 2004 (Courtesy Prof. Mitsuyasu Hasebe) was the wild-type (WT) *Physcomitrium (Physcomitrella) patens* genetic material we used. All plant materials were grown on BCD medium supplemented with 5 mM ammonium tartrate and 1 mM CaCl_2_. Plants were grown at 25 ℃ under 16 h light photoperiod per 8 h dark photoperiod with light intensity 60–80 µmol photons m^−2^ s^−1^.

### 4.3. Protoplast Transformation

Protoplast of *P. patens* were prepared from protonema which were continuously disrupted with an electric stirrer every week. The overexpression vector pPOG1-*A* was linearized using the restriction enzyme *Mss*I (Thermo Fisher Scientific, Waltham, MA, USA). The transformants were obtained by transferring the linearized vector into wild-type using polyethylene glycol (PEG)–mediated protoplast transformation [63].

### 4.4. PCR and Real-Time qRT-PCR Characterization of Overexpression Plants

Stable transgenetic lines were identified by PCR and qRT-PCR at both DNA and RNA level. The transformants were initially screened at the DNA level using primers “F” and “R”. Total RNA from *P. patens* tissues was extracted using TRIzol (Invitrogen, Carlsbad, CA, USA) according to the instructions. cDNA synthesis was performed using a PrimeScript™ RT reagent Kit with gDNA Eraser (Takara, Dalian, China) according to the manufacturer’s instructions. qRT-PCR was performed using SYBR Premix Ex Taq II (Takara, Dalian, China) and carried out on Bio-Rad CFX96. The primer sequences for PCR and qRT-PCR are listed in Appendix A.

### 4.5. Analysis of Gene Expression Patterns and Analysis of Expression by Stress Treatment

Expression pattern data were retrieved and visualized from the public database *Physcomitrella* eFP Browser of BAR (http://bar.utoronto.ca/, accessed on 19 February 2022). 

### 4.6. Observation of Cell Ultrastructure

Leaves were cut into 1 mm^2^ with a blade and immediately fixed with 3% glutaraldehyde overnight at 4 °C. Rinse the material by adding 0.1 M phosphate buffer (PBS buffer, pH 7.2) every 30 min for 3 times. Subsequently, 1% osmium tetroxide was added for sample fixation at 4 °C for 2 h. Continue rinsing three times for 20 min each. The materials were continuous rinsed for 3 times and dehydrated in a serial ethanol gradient. After embedded in Epon 812 resin, materials were sectioned and stained with 2% uranyl acetate and lead citrate under EM UC7 ultramicrotome (Leica, Weztlar, Germany). The cellular ultrastructure was observed under JEM-1400Plus transmission electron microscopy (JEOL, Tokyo, Japan).

### 4.7. Observation of Cell Surface Structure

Leaves of the material were cut into 9 mm^2^ with a blade and immediately fixed with 3% glutaraldehyde overnight at 4 °C. The rinsing, fixation, and dehydration procedures are the same as previous mentioned. Samples were dried with a CO_2_ critical-point drier. The dried materials were mounted on aluminium stubs using tweezer and sputter-coated with gold. Cell surface structure was observed by Zeiss Sigma 300 scanning electron microscopy (Zeiss, Oberkochen, Germany).

### 4.8. Nile Red Staining

Gametophytes grown for 30 days were taken and placed in a certain volume of Nile red working solution for staining in dark for 5 mins as previously described, with appropriate modifications [64]. Lipid droplets were observed under Olympus FV1000 confocal microscope (Olympus, Tokyo, Japan). The emission filters for Nile red were 493–636 nm.

### 4.9. Heat Stress Assay

Thirty-days old gametophytes of *P. patens* were used to perform a heat stress assay. The wild-type plants and overexpression plants were treated at 40 °C for 18 h, followed by a recovery at 25 °C for 4–5 days.

### 4.10. Measurement of Chlorophyll Fluorescence (Fv/Fm), Non-Photochemical Energy Dissipation (NPQ), Electron Transport Rate (ETR), and Pigments Content

The plants to be tested were placed in the dark for 30 min. Relevant photosynthetic parameters were determined using IMAGING-PAM (Walz, Effeltrich, Germany) and the Imaging Win software (Walz, Effeltrich, Germany) as described previously [65]. Pigment content was determined using the N, N-dimethylformamide (DMF) method, as described previously [65].

### 4.11. Bioinformatics and Data Analysis

Wild-type and *A*-OE plants before and after heat treatment were used to extract RNA. Samples for RNA sequencing (RNA-seq) were treated as previous description [66]. Differentially expressed genes (DEGs) between two samples were identified with the criteria of “Fold change > 1.5” and “*p* value < 0.05”. Differentially expressed genes were visualized with TBtools [67]. The dehydrated transcriptome data used in this study were obtained from Dong et al. [46].

### 4.12. Statistical Analysis

All experiments were performed with three biological replicates. Student’s *t*-test was used for hypothesis testing in statistics between two samples. Significant differences were defined and indicated by asterisks *, **, and ***, corresponding to *p*-values < 0.05, <0.01, and <0.001, respectively.

## 5. Conclusions

In conclusion, we revealed that *PpRNH1A* not only participates in the regulation of growth development of *P. patens* plants, it also plays a crucial role in plant response and tolerance to abiotic stresses such as heat, possibly by regulating the expression of heat-related genes and causing the abnormal accumulation and the mobilization of lipid droplets in the cytosol. Our data highlights the important role played by *PpRNH1A* in plant heat stress response, providing a novel insight into the function of RNase H1s.

## Figures and Tables

**Figure 1 ijms-23-09270-f001:**
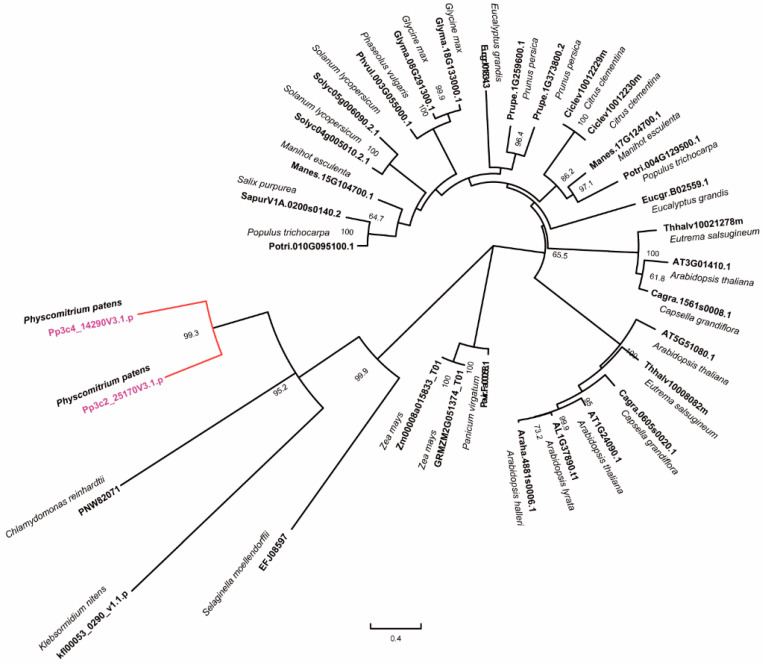
Phylogenetic analyses of RNase H1 in different species. The tree was constructed with MEGA-X using the maximum likelihood (ML) method with 1000 bootstrap replicates. The numbers above the branches represent the bootstrap support values (>50%) from 1000 replications.

**Figure 2 ijms-23-09270-f002:**
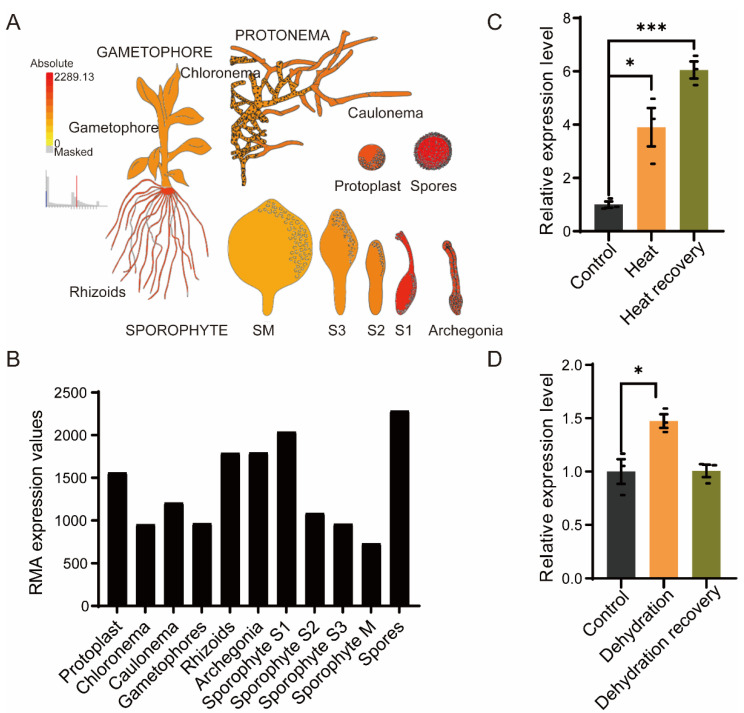
Gene expression patterns of *PpRNH1A* throughout the life cycle of *P. patens* and the expression levels under different stresses. (**A**) Visualization of the expression of *PpRNH1A* at different tissues during *P. patens* growth and development. (**B**) The RMA expression values of *PpRNH1A* at different periods of growth and development. Data were downloaded from *Physcomitrella* eFP Browser. (**C**) Gene expression levels of *PpRNH1A* under heat stress and after heat recovery. (**D**) Gene expression levels of *PpRNH1A* under dehydrating stress and after recovery. For (**C**,**D**), data are presented as means ± SEM of three replicates; *t*-test was used for statistics; asterisks indicate the significant difference between treatment group and the control group, * *p* < 0.05, *** *p* < 0.001.

**Figure 3 ijms-23-09270-f003:**
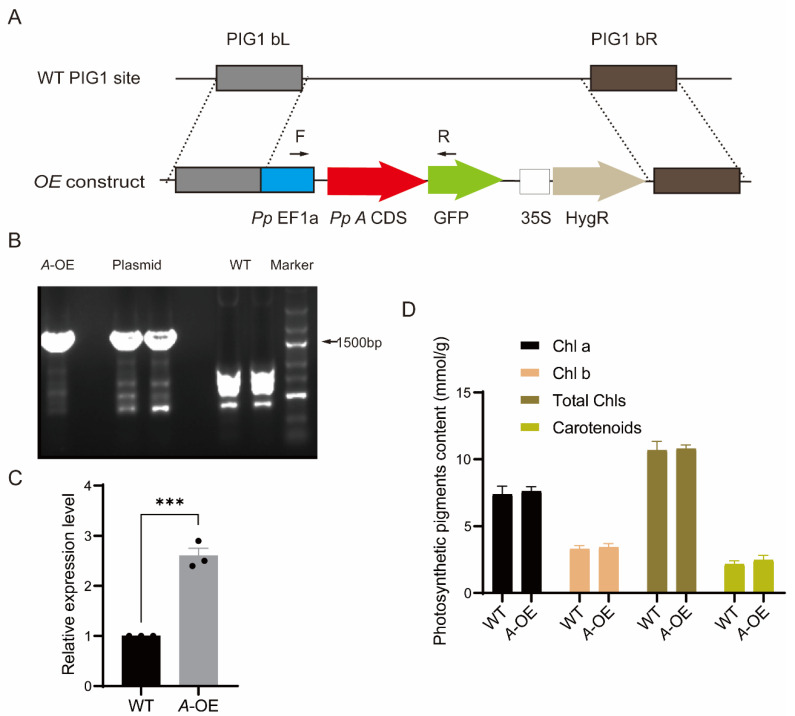
Stable *PpRNH1A*-overexpression lines in *P. patens*. (**A**) Schematic diagram of pPOG1-*A* construct. P, plasmid; PIG1 bL, left targeting site; PIG1 bR, right targeting site; *Pp*EF1a, the EF1a promoter from *P. patens*; HygR, hygromycin selectable marker cassette; GFP, green fluorescent protein. (**B**) Positive *PpRNH1A*-overexpression (*A*-OE) plants were confirmed at DNA level. WT, wild type; *A*-OE, *PpRNH1A*-overexpressing plant. (**C**) Identification of *A*-OE plant by qRT-PCR at RNA level. (**D**) Pigment contents in WT and *A*-OE under normal growth conditions. Chl a: chlorophyll a; Chl b: chlorophyll b; Total Chls: Total chlorophylls. For (**C**,**D**), data are presented as means ± SEM of three replicates; *t*-test was used for statistics; asterisks indicate the significant difference between WT and *A*-OE, *** *p* < 0.001.

**Figure 4 ijms-23-09270-f004:**
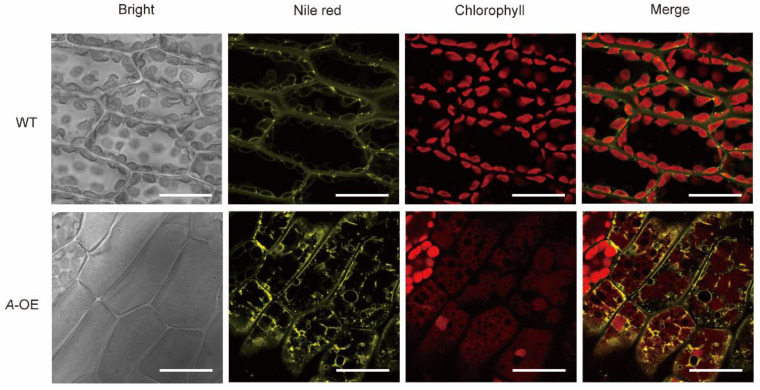
Accumulation of lipid droplets in WT and *A*-OE under normal growth conditions. Scale bar = 20 μm.

**Figure 5 ijms-23-09270-f005:**
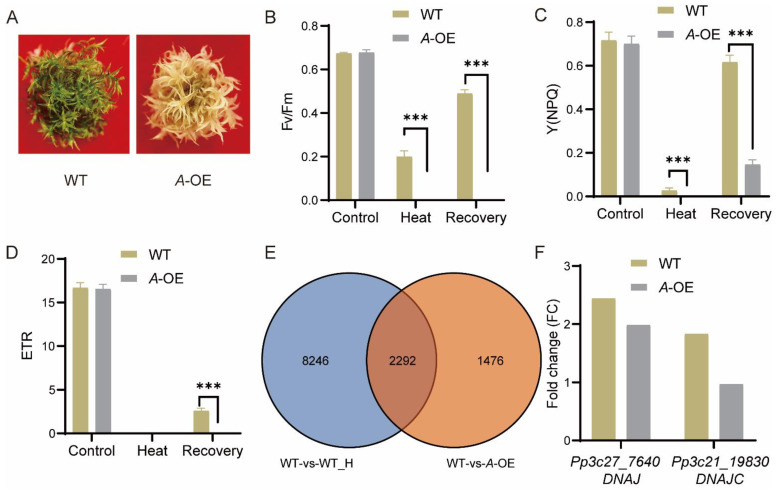
Response of *A*-OE plants to heat stress (HS) treatment. (**A**) *A*-OE plants were more sensitive to heat stress. (**B**–**D**) Chlorophyll fluorescence (Fv/Fm) (**B**), non-photochemical energy dissipation (NPQ) (**C**), and electron transport rate (ETR) (**D**) of WT and *A*-OE before HS, after HS, and after recovery from the heat stress. Data are presented as means ± SEM of three replicates; *t*-test was used for statistics; asterisks indicate the significant difference between WT and *A*-OE, *** *p* < 0.001. (**E**) Venn diagram shows the overlapping between HS-regulated DEGs in WT (WT versus WT under heat stress) and *PpRNH1A*-regulated DEGs (WT versus *A*-OE under normal growth conditions). (**F**) Fold change comparison of the expression levels of *DNAJ* and *DNAJC* under HS and normal conditions. Each bar represents the ratio of the average expression value of three biological replicates of HS (after heat stress) vs. control (before heat stress).

## Data Availability

The RNA-seq data reported in this paper have been uploaded in National Genomics Data Center (accession no. PRJCA011244).

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
