# Peer review of "Regulation of Heat Stress in Physcomitrium (Physcomitrella) patens Provides Novel Insight into the Functions of Plant RNase H1s"

_ijms, 2022, doi:10.3390/ijms23169270_

Round 1
Reviewer 1 Report
The work appears very well structured. The topic is certainly very interesting and very topical, and makes a significant contribution to the study of the behavior of species to environmental stresses. The methodology adopted appears to be very accurate and the observations resulting from the experimentation are interesting. The bibliography is complete and exhaustive. English is also of good quality.
Author Response
Jul 19, 2022
Dear reviewer:
Thank you very much for your positive comments concerning our manuscript entitled "Regulation of heat stress in Physcomitrium (Physcomitrella) patens provides novel insight into the functions of plant RNase H1s".
Thank you and best wishes,
Li Liu
Reviewer 2 Report
My expertise lies in cytology (light microscopy, TEM,SEM) and development including Physcomitrium. I cannot comment on the molecular aspects of this study.
The authors seem to have a very poor understanding of bryophyte fine structure.
Mosses do not have oil bodies. These are membrane-bound bodies unique to liverworts. This paper illustrates changes in cytoplasmic lipid droplets and plastoglobuli inside the chloroplasts. They need to look up the literature on these.
The TEMS are of poor quality with poor contrast and are only marginally fit for publication. The walls in A WT are badly creased.
The SEM is incomprehensible. C A-OE needs light microscope images to show the so-called abnormalities.
D looks like spores of some kind not wax particles. In any case these would likely have been removed by the ethanol in the preparation process. I have made extensive LM, TEM and SEM (including cryo) observations on Physcomitrium and numerous other mosses and never seen wax particles these.
The TEM and SEM need redoing and the manuscript revised accordingly.
Author Response
Jun 19, 2022
Dear reviewer:
Thank you very much for reading and commenting on our paper entitled "Regulation of heat stress in Physcomitrium (Physcomitrella) patens provides novel insight into the functions of plant RNase H1s". We have revised the manuscript according to your suggestions. Point-by-point responses were listed in the attachment.
Thank you and best wishes,
Li Liu

Round 2
Reviewer 2 Report
I reviewed the original version of this paper.
I am very disappointed by the authors’ responses to my comments, and the work still requires revision of the cytology to render it suitable for publication.
FIG 4 A WT is unfit for publication and the legends need much more detail. A A-OE. 4A and B show two kinds of lipid droplet : large ones in the cytoplasm and plastoglobuli. Did the frequency of the latter change under stress as has been reported previously? Nowhere are these mentioned . Line 173. This is simply poor fixation.
4C remains incomprehensible and 4D shows spores NOT waxes. I reiterate, waxes like this have never been described in bryophytes. Crystals are NOT spherical. The best solution is to omit all reference to surfaces and waxes in the cytology
L55 the authors have failed to clarify what is meant by oil bodies (they might look up references on oleosomes). The structures they are describing are NOT the same as liverwort oil bodies which are membrane-bound. Past terminology of oil bodies has been vague but that is no reason for getting it wrong here. Why not simply the call the cytoplasmic lipid droplets or oleosomes. Somewhere after line 55 they must state that the structures they are describing are NOT the same as liverwort oil bodies.
Please see my comments in bold in the word file below of the authors’ responses
Response to Reviewer 2 Comments
Point 1: Mosses do not have oil bodies. These are membrane-bound bodies unique to liverworts. This paper illustrates changes in cytoplasmic lipid droplets and plastoglobuli inside the chloroplasts. They need to look up the literature on these.
Response 1: We appreciate your raising this issue. Indeed, the term “oil body” was described in liverworts, with many studies revealed the ultrastructure of oil body, the size, development, composition, and functions of oil bodies in liverworts [1-11]. Recently, it was also found that oil body formation in Marchantia polymorpha is controlled by MpC1HDZ, and its potential roles in protecting plants from abiotic and biotic stresses [11].Irrelevant as liverwort oil bodies are not what is being described in Physco However, the term “oil body” have also been wildly used in other plant species with a broader range of definition including lipid droplets and lipid bodies, and they were commonly found in different plant tissues including pollens, leaves, fruits, and especially in plant seeds [12, 13, 14]. Furthermore, oil bodies were also found and described in physcomitrium patens, which exhibit the features and function of oil bodies in mosses [15, 16]. Therefore, we think the term “oil body” could be used in this manuscript. To address the concern, we have made modifications in the introduction of our manuscript to induce “oil body”. From line 65 to 70, we added a description of oil body in plants: "Oil body (or so-called lipid droplets or lipid bodies in some cases) is a special oganelle capable to store lipids deposited in eukaryotes, which is surrounded by a monolayer of membrane lipids [18, 19]. Oil body is widely found and explored in various kind of tissues such as seeds, pollen, and leaves in different plant species [20, 21, 22, 23]. Abundant Oil bodies were found to be present in dehydrated spores and in photosynthetic gametophyte in P. patens [23]."
Point 2: The TEMS are of poor quality with poor contrast and are only marginally fit for publication. The walls in A WT are badly creased.
Response 2: Thank you for pointing out this issue in Figure 4A. We feel very sorry for the creased problem in WT, maybe because the sample might not be well handled when we did the TEM. However, we think it doesn't affect the ultrastructure phenotype of the PpRNH1A overexpression transgenic line that we described in our results.
Point 3: The SEM is incomprehensible. C A-OE needs light microscope images to show the so-called abnormalities.
Response 3: Thank you for pointing out this issue in our figure 4C. We are sorry that we did not use the light microscope to show the abnormality of A-OE phenotype. Instead, we used the SEM since SEM was reported to be used to observe the structure of gametophores, phyllid apices, and epicuticular waxes of moss and liverwort [17, 18]. To make our figure more comprehensible and address this issue, we tried to use arrows to point out the protrusions and abnormalities on the surface of the A-OE plants in Figure 4. 2 These SEMS remain unpublishable
Point 4: D looks like spores of some kind not wax particles. In any case these would likely have been removed by the ethanol in the preparation process. I have made extensive LM, TEM, and SEM (including cryo) observations on Physcomitrium and numerous other mosses and never seen wax particles these. The TEM and SEM need redoing and the manuscript revised accordingly.
Response 4: Thanks for the comments on figure 4D. Yes, we also do not think that they are spores although they look very like. Maybe “wax crystals” is a more appropriate description. Wax crystals are considered to be a major component of cuticle and could protect plant from water loss. According to previous reports in mosses, wax crystals could be detected and observed using grazing incidence wide angle X-ray scattering (GIWAXS) and SEM [19, 15]. Therefore, we have changed the "wax particles" into "wax crystals" throughout the manuscript. Omit all references to waxes as these are spores
Point 5: The TEM and SEM need redoing and the manuscript revised accordingly.
Response 5: Many thanks for your suggestion! We agree that the quality of the manuscript will be improved if we could redo the TEM and SEM. However, it will take a long time and heavy cost for us to redo these experiments to get better quality figures. This is not true. Since the authors already have embedded material all they need to do is cut a few more sections and look at these under the TEM-2 days work at most. As it stands it would be better to leave out ALL THE EM. Since the TEM and SEM in our Figure 4 has not much effect on our conclusion of the huge difference between the WT and the PpRNH1A overexpression transgenic line, although not perfect, we are sorry that we might not be able to re-do the experiments as suggested. However, we really appreciate the comments and have revised manuscript extensively. Hope the reviewer and the editor could understand and accept our revisions.
I reviewed the original version of this paper.
I am very disappointed by the authors’ responses to my comments, and the work still requires revision of the cytology to render it suitable for publication.
FIG 4 A WT is unfit for publication and the legends need much more detail. A A-OE. 4A and B show two kinds of lipid droplet : large ones in the cytoplasm and plastoglobuli. Did the frequency of the latter change under stress as has been reported previously? Nowhere are these mentioned . Line 173. This is simply poor fixation.
4C remains incomprehensible and 4D shows spores NOT waxes. I reiterate, waxes like this have never been described in bryophytes. Crystals are NOT spherical. The best solution is to omit all reference to surfaces and waxes in the cytology
L55 the authors have failed to clarify what is meant by oil bodies (they might look up references on oleosomes). The structures they are describing are NOT the same as liverwort oil bodies which are membrane-bound. Past terminology of oil bodies has been vague but that is no reason for getting it wrong here. Why not simply the call the cytoplasmic lipid droplets or oleosomes. Somewhere after line 55 they must state that the structures they are describing are NOT the same as liverwort oil bodies.
Please see my comments in bold in the word file below of the authors’ responses
Response to Reviewer 2 Comments
Point 1: Mosses do not have oil bodies. These are membrane-bound bodies unique to liverworts. This paper illustrates changes in cytoplasmic lipid droplets and plastoglobuli inside the chloroplasts. They need to look up the literature on these.
Response 1: We appreciate your raising this issue. Indeed, the term “oil body” was described in liverworts, with many studies revealed the ultrastructure of oil body, the size, development, composition, and functions of oil bodies in liverworts [1-11]. Recently, it was also found that oil body formation in Marchantia polymorpha is controlled by MpC1HDZ, and its potential roles in protecting plants from abiotic and biotic stresses [11].Irrelevant as liverwort oil bodies are not what is being described in Physco However, the term “oil body” have also been wildly used in other plant species with a broader range of definition including lipid droplets and lipid bodies, and they were commonly found in different plant tissues including pollens, leaves, fruits, and especially in plant seeds [12, 13, 14]. Furthermore, oil bodies were also found and described in physcomitrium patens, which exhibit the features and function of oil bodies in mosses [15, 16]. Therefore, we think the term “oil body” could be used in this manuscript. To address the concern, we have made modifications in the introduction of our manuscript to induce “oil body”. From line 65 to 70, we added a description of oil body in plants: "Oil body (or so-called lipid droplets or lipid bodies in some cases) is a special oganelle capable to store lipids deposited in eukaryotes, which is surrounded by a monolayer of membrane lipids [18, 19]. Oil body is widely found and explored in various kind of tissues such as seeds, pollen, and leaves in different plant species [20, 21, 22, 23]. Abundant Oil bodies were found to be present in dehydrated spores and in photosynthetic gametophyte in P. patens [23]."
Point 2: The TEMS are of poor quality with poor contrast and are only marginally fit for publication. The walls in A WT are badly creased.
Response 2: Thank you for pointing out this issue in Figure 4A. We feel very sorry for the creased problem in WT, maybe because the sample might not be well handled when we did the TEM. However, we think it doesn't affect the ultrastructure phenotype of the PpRNH1A overexpression transgenic line that we described in our results.
Point 3: The SEM is incomprehensible. C A-OE needs light microscope images to show the so-called abnormalities.
Response 3: Thank you for pointing out this issue in our figure 4C. We are sorry that we did not use the light microscope to show the abnormality of A-OE phenotype. Instead, we used the SEM since SEM was reported to be used to observe the structure of gametophores, phyllid apices, and epicuticular waxes of moss and liverwort [17, 18]. To make our figure more comprehensible and address this issue, we tried to use arrows to point out the protrusions and abnormalities on the surface of the A-OE plants in Figure 4. 2 These SEMS remain unpublishable
Point 4: D looks like spores of some kind not wax particles. In any case these would likely have been removed by the ethanol in the preparation process. I have made extensive LM, TEM, and SEM (including cryo) observations on Physcomitrium and numerous other mosses and never seen wax particles these. The TEM and SEM need redoing and the manuscript revised accordingly.
Response 4: Thanks for the comments on figure 4D. Yes, we also do not think that they are spores although they look very like. Maybe “wax crystals” is a more appropriate description. Wax crystals are considered to be a major component of cuticle and could protect plant from water loss. According to previous reports in mosses, wax crystals could be detected and observed using grazing incidence wide angle X-ray scattering (GIWAXS) and SEM [19, 15]. Therefore, we have changed the "wax particles" into "wax crystals" throughout the manuscript. Omit all references to waxes as these are spores
Point 5: The TEM and SEM need redoing and the manuscript revised accordingly.
Response 5: Many thanks for your suggestion! We agree that the quality of the manuscript will be improved if we could redo the TEM and SEM. However, it will take a long time and heavy cost for us to redo these experiments to get better quality figures. This is not true. Since the authors already have embedded material all they need to do is cut a few more sections and look at these under the TEM-2 days work at most. As it stands it would be better to leave out ALL THE EM. Since the TEM and SEM in our Figure 4 has not much effect on our conclusion of the huge difference between the WT and the PpRNH1A overexpression transgenic line, although not perfect, we are sorry that we might not be able to re-do the experiments as suggested. However, we really appreciate the comments and have revised manuscript extensively. Hope the reviewer and the editor could understand and accept our revisions.
Author Response
Aug 2rd, 2022
Dear reviewer:
Thank you very much for reading and commenting on our paper entitled "Regulation of heat stress in Physcomitrium (Physcomitrella) patens provides novel insight into the functions of plant RNase H1s" again. We have revised the manuscript according to your suggestions. Point-by-point responses were listed in the attachment.

Round 3
Reviewer 2 Report
Much improved on the second version.
All my major concerns have now been dealt with